# Influence of the Catalyst Layer Structure Formed by Inkjet Coating Printer on PEFC Performance

**DOI:** 10.3390/polym13060899

**Published:** 2021-03-15

**Authors:** Yushi Tamaki, Kimihiko Sugiura

**Affiliations:** Department of Technological Systems, Mechanical Engineering Course, Osaka Prefecture University College of Technology, 26-12 Saiwai-cho, Neyagawa, Osaka 572-8572, Japan; sugiura@osaka-pct.ac.jp

**Keywords:** PEFC, catalyst ink, catalyst layer, inkjet

## Abstract

In this study, we investigated the influence of the Catalyst-Layer (CL) structure on Polymer Electrolyte Fuel Cell (PEFC) performance using an inkjet coating printer, and we especially focused on the CL thickness and the electrode area. In order to evaluate the influence of CL thickness, we prepared four Membrane Electrode Assemblies (MEAs), which have one, four, five and six CLs, respectively, and evaluated it by an overpotential analysis. As a result, the overpotentials of an activation and a diffusion increased with the increase of thickness of CL. From Energy Dispersive X-ray spectroscopy (EDX) analysis, because platinum twines most ionomers and precipitates, the CL separates into a layer of platinum with a big grain aggregate ionomer and the mixing layer of platinum and ionomer during the catalyst ink drying process. Consequently, the activation overpotential increased because the three-phase interface was not able to be formed sufficiently. The gas diffusivity of the multilayer catalyst electrode was worse than that of a single layer MEA. The influence of the electrode area was examined by two MEAs with 1 and 9 cm^2^ of electrode area. As a result, the diffusion overpotential of 9 cm^2^ MEA was worse than 1 cm^2^ MEA. The generated condensate was multiplied and moved to the downstream side, and thereafter it caused the flooding/plugging phenomena.

## 1. Introduction

Our society has faced various environmental issues represented by the global-warming phenomenon, and we need to reduce the emission of greenhouse gas to mitigate such issues. However, because the consumption of fossil fuels has accelerated due to rapid development of developing countries, advanced countries have the obligation to accelerate technology development for everyone to be able to obtain affordable power by clean energy. In this situation, hydrogen energy is focused on as an alternative energy, and the popularization of polymer electrolyte fuel cells (PEFCs) was begun. By 2019, 200,000 units, or more, of Ene-farm were sold as a domestic cogeneration system, and about 2926 fuel cell vehicles (FCVs) were sold in Japan [1].

However, the coverage of PEFC is quite different from the road map of the Ministry of Economy, Trade and Industry. This principal factor is that the system cost is too high. For example, the payback time of Ene-farms is five years or more, and the price difference between an FCV and a hybrid vehicle is about 30,000 dollars [1]. Therefore, reduction of the system cost is necessary for further popularization of PEFCs. A lot of researchers have been studying how to enhance the performance of PEFC components such as polymer electrolyte membranes (PEM) [2,3] or catalysts. Especially, the reduction of the amount of Pt catalyst is important because it accounts for about 46% of the system cost. In previous research, Pt alloying catalysts [4,5,6,7,8] and Pt core-shell structural catalysts [9,10,11,12,13] were studied. Furthermore, research is being conducted on new catalysts to completely replace Pt catalysts [14,15,16]. However, almost all these studies are currently in the research or project stages, and it take a long time before research results are applied to products. Therefore, it is important to promote Pt reduction by improving commercialized technology. From this point of view, electrode manufacturing methods such as the decal transfer method, the splay deposition method, and an ultrasonic-splay method have been studied [17,18,19].

As one of the methods, applying Ink Jet Coating method (IJC method) has been studied recently [20,21,22]. The IJC method exhales ink from the nozzle head as a micro droplet and forms a layer composed of numerous droplets. Because IJC method can exhale ink in a precise manner, we consider that CL cannot only be formed with an amount of Pt that is less than that of a conventional Membrane Electrode Assembly (MEA), but also a new structure that cannot be manufactured by conventional methods. We compared MEA made by the IJC method (MEA-IJC) and MEA made by a doctor blade (MEA-DB) in prior work. As a result, the performance of the cell with MEA-IJC was superior to MEA-DB though the Pt amount of MEA-IJC was 36.6 wt.% lower than MEA-DB [23]. Consequently, we concluded that the reduction in thickness of CL improved cell performance. However, the relationship between the CL structure and cell performance was hardly elucidated. Moreover, because we have conducted experiments on only 1 cm^2^ electrodes in the prior work, the influence of an electrode area on MEA performance using IJC method has not been examined yet. It is necessary to elucidate these effects to manufacture a CL with complex structure. Therefore, in this study, we aimed to elucidate the influence of the CL thickness and the influence of an electrode area on cell performance using the IJC method.

## 2. Materials and Methods

### 2.1. Settings on Inkjet Printer and Preparation of Catalyst Ink for Inkjet Printer

We used a Microjet Labojet-500, a piezo-electric type inkjet printer, and the nozzle head was for high-viscosity and high-surface tension (IJHD-1000, Microjet, Nagano, Japan). The droplet condition was adjusted by the nozzle head driving parameter as shown in Table 1.

The catalyst ink was a blend of 0.50 g of platinum catalyst (TEC10V40E 38.0 wt.% PT/CB, Tanaka Kikinzoku Kogyo, Tokyo, Japan), 7.0 g of purified water, 2.75 g of ethanol and 1.53 g of Nafion solution (DEC2021CS 20% Nafion dispersion, Wako Pure Chemical Industries, Osaka, Japan) using a screw vial, and stirred by a rotor for 12 h. Finally, as for the catalyst ink, distributed processing was applied for 180 s by the homogenizer at power of 20 W, and the defoaming processing was also applied for 3 min by the vacuum stirring mill.

### 2.2. Application Method by Inkjet Printer and Preparation of MEA

The catalyst ink was exhaled on a Micro Porous Layer (MPL) side of a Gas Diffusion Layer (GDL) (GDL29BC, SGL Group) every 100 μm according to *x*-axis and *y*-axis by in-line arrangement as shown on Figure 1. To elucidate the influence of the thickness on cell performance, we prepared four MEAs which have one, four, five and six CLs for each side, respectively, as shown in Figure 2a–d. Additionally, to elucidate the influence of the electrode area on cell performance, we prepared two MEAs which have 1 cm^2^ and 9 cm^2^ of electrode area. Here, the catalyst layer was one layer as shown in Figure 2e. The thickness of one CL is about 5 μm as shown in Scanning Electron Microscope image (SEM) of Figure 3a and the thickness increases each time the CL is laminated, as shown in Figure 3b–d. We consider that the reason why the CL thickness is not increasing proportionally to the number of application times is that the catalyst ink flowed into concave areas on the CL surface. After the ink dried, a polymer electrolyte membrane (Nafion NRE-212, Sigma-Aldrich Japan, Tokyo, Japan) was placed between two GDLs to which the catalyst layer was applied, and it was pressed for five minutes with a hot press at 123 °C and 8 MPa.

### 2.3. Experimental Apparatus and Method

Although an effective electrode area of the 1 cm^2^ MEA and 9 cm^2^ MEA is different, all MEAs were evaluated using a common single cell with 3 × 3 cm^2^ electrode area. The gas channel was a single serpentine channel which had 1.8 mm^W^ × 1.2 mm^D^, and the width of a rib was 1 mm. Here, because 1 cm^2^ of MEA contacted with 3 gas channels and 4 ribs, its contact area between MEA and rib was only 0.4 cm^2^ though that of the 9 cm^2^ MEA was 2.88 cm^2^. Because 1 cm^2^ of MEA only had the membrane in the parts besides the 1 cm^2^ part of the midrange, the gas seal was applied by supplementing the thickness of GDL in the part which had this film with PTFE sheet.

Cell performance was evaluated by I-V characteristics curve and Cole–Cole plot using a fuel cell impedance meter (KFM2030, KIKUSUI). Table 2 shows the experimental condition and the humidifying condition. Because the cell temperature and each humidifier temperature was 80 °C, the relative humidity in the cell was 100%. The fuel gas and the oxidant gas utilizations were U_fuel_/U_ox_ = 70%/40% @0.4 A/cm^2^ defined as the 9 cm^2^ of the single cell.

### 2.4. A Method of Overpotential Analysis

Generally, the deterioration factors of cell voltage on PEFC are classified as overpotentials of resistance, an activation and a diffusion. In this study, we referred to the protocol conducted by Vielstich et al. as the overpotential analysis [24].

(1)We assumed the relationship as Equation (1): (1)$$V_{th}=V_{act}+V_{ir}+V_{diff}+V_{cell}$$
where *V_th_* is the theoretical open circuit voltage in the PEFC (1.23 V), *V_act_* is an activation over potential, *V_ir_* is a resistance overpotential, *V_diff_* is a diffusion over potential and *V_cell_* is a cell voltage, respectively.(2)The resistance over potential *V_ir_* is derived using Equation (2) from the resistance *R* measured by the measurement of I-V performance: (2)$$V_{ir}=ir=\frac{I}{A_{e}}\times \left(R\times A_{c}\right)$$
where *i* A/cm^2^ is the current density, *r* Ωcm^2^ is an area resistance, *I* A is a current, *A_e_* cm^2^ is an electrode area, *R* Ω is a measured resistance and *A_c_* cm^2^ is a contact area, respectively.(3)I-V performance was converted to an IR-free curve, thereafter the I-V curve was rearranged to Tafel plot. Here, an activation overpotential (*V_act_*) was derived by the difference between the regression curve of the voltage of the Tafel plot and the theoretical voltage in the low current density region.(4)A diffusion overpotential (*V_diff_*) was defined by the value in which the above-mentioned regression curve of voltage and IR-free are subtracted from the theoretical open circuit voltage.

## 3. Results and Discussion

### 3.1. Influence of the CL Thickness on Cell Performance

Figure 4 and Figure 5 show the result of overpotential analysis of each thickness on 1 cm^2^ of MEA and the result of Cole–Cole plot, respectively. The maximum current density of MEA1CL is 0.85 A/cm^2^, which is the highest as shown in Figure 4a. The impedance also increases with the increase of the thickness of CL as shown in Figure 5. The overpotentials of the activation and the diffusion were greatly responsible for the thickness of CL as shown in Figure 4a–d. Although the activation overpotential of MEA4CLs is better than that of MEA1CL, that of MEA5CLs and MEA6CLs are worse as shown in Figure 4b. Because the CL thickness of these three MEAs is thicker than MEA1CL, and the amount of platinum is also larger, the activation overpotential of these MEAs should be improved compared with that of MEA1CL. Therefore, it is suggested that the additional Pt catalyst was not utilized efficiently. Because we consider that this is due to to the proton conductivity, the cross section of the MEA was analyzed by energy dispersive X-ray spectroscopy (EDX). Figure 6 shows the result of Pt and S mapping analysis on the cross-section of MEA5CLs using EDX. Here, Pt and S represent the Pt catalyst and the ionomer composed of sulfonic, respectively, in this SEM image. The peak intensities of Pt and S are at the interface between CL and GDL (about 50 µm from right edge of the SEM image), and the peak of S decreases while moving right (membrane side). Therefore, we consider that because platinum twines most ionomers and precipitated, the CL separated into a layer of only platinum and the mixing layer of platinum and ionomer during the catalyst ink drying process. Here, some researchers reported that ionomers indicate strong agglutination with Pt/CB in water-rich solution by hydrophobic interaction [25,26]. Kumano et al. reported that although 23 wt.% of ionomers was adsorbed on the Pt/CB surface in the case of ink components shown on Table 3, the ionomer of 77% was dispersed to the entire CL [25]. Although we considered that the same phenomenon occurred in this study because the water and organic solvent ratio on our catalyst ink is approximate to the reference as shown on Table 3, the ionomer hardly existed on the membrane side as shown in Figure 6. Therefore, we observed the surface of each CL by SEM as shown in Figure 7. As a result, a lot of grain aggregates of about 10 µm are scattered on the surface of each CL. Platinum and the ionomer that was not able to be precipitated became a big grain aggregate by cohesion, and were scattered in the CL. Because the Pt catalyst does not touch the ionomer, the Pt catalyst was not able to obtain a proton, and it was not able to cause the cathode reaction. Consequently, because this tendency grows as the catalyst thickness increases, the activation overpotential of MEA with thick CL has grown as shown in Figure 8. Therefore, the dispersion of ionomer should be improved when the CL is applied by IJC.

The diffusion overpotential of MEA1CL, -4CLs, -5CLs and -6CLs under 0.25 A/cm^2^ of the current density are 0.0466 V, 0.0562 V, 0.159 V and 0.243 V, respectively, as shown in Figure 4d. It is clear that the diffusion overpotential increases with the increase of CL thickness (number of CL). The diffusion resistance of oxygen generally grows as the thickness of the CL increases. This tendency grows with the increase of the current density. As shown in Figure 7, the crack on the surface of CL grows noticeably as the thickness of CL increases. The crack width of MEA6CLs reaches about 100 μm. Almost all the researchers reported that the crack of the CL depraves the gas diffusion and the reactivity [27,28], though some researchers reported that a specific crack structure improved gas diffusivity [29]. Therefore, because the oxygen diffusion was disturbed with water generated by the cell reaction that was accumulated by these cracks, the diffusion overpotential rose and consequently the deterioration was caused.

In addition, from the perspective of cell durability, several researchers reported that cracks in the CL reduces cell durability [30,31,32]. Consequently, it is desirable to study the ink exhaling method that suppresses the occurrence of cracks.

### 3.2. Influence of the Electrode Area on Cell Performance

Figure 9 and Figure 10 show the result of overpotential analysis and the Cole–Cole plot of each electrode area, respectively. The maximum current density of 9 cm^2^ MEA is 0.33 A/cm^2^, and is 0.52 A/cm^2^ smaller than 1 cm^2^ MEA, as shown in Figure 9a. Although the contact area between MEA and rib of 9 cm^2^ MEA is 7.2 times that of 1 cm^2^ MEA, both resistance overpotentials are almost the same as shown in Figure 9c. Moreover, because the three-phase interface that is a reaction field does not change even if the electrode area changes, both activation overpotentials are also almost the same as shown in Figure 9b. Therefore, it is clear that the diffusion overpotential is a deteriorated principal factor as shown in Figure 9b–d and Figure 10. The difference in the diffusion overpotential originates from the electrode area because the method of application, the catalyst ink composition, and experimental conditions, etc., other than the electrode area, are the same. We consider that MEA of 9 cm^2^ is inferior to MEA of 1 cm^2^ by drainage according to the extension of the gas channel. Furthermore, 1 cm^2^ of MEA was located in the midrange of the separator of 9 cm^2^ as shown in Figure 11a, and it was contacted with 3 gas channels and 4 ribs as mentioned above. Therefore, although the area that MEA of 1 cm^2^ and 9 cm^2^ touch the channel are 0.54 cm^2^ and 6.12 cm^2^, respectively, as shown in Figure 11, both percentages of the contact area with the channel are almost the same. The supply gas amount of both MEAs is also the same but the oxygen consumption and the vapor quantity produced by the cell reaction according to the electric current density differ nine-fold. Although the relative humidity of 1 cm^2^ MEA is 102%, that of 9 cm^2^ MEA is 114%. Therefore, the condensate is accumulated toward the gas outlet, and causes the plugging by which the condensate blockades the gas passage in the vicinity of the gas outlet. Consequently, the diffusion overpotential of 9 cm^2^ MEA increases.

## 4. Conclusions

In this study, we investigated the influence of the thickness of CL (number of CLs) and the influence of electrode area on the cell performance.

In the influence of the thickness of CL on the cell performance, we confirmed that the activation overpotential and the diffusion overpotential were increased with the increase of thickness of CL (number of CLs) from the overpotential analysis. From the EDX analysis and SEM image, we understood that platinum twines about 23 wt.% of ionomers and precipitated, and the platinum and the ionomer that were not able to be the precipitated became a big grain aggregate by cohesion and were scattered in the CL during the catalyst ink drying process. Moreover, the crack on the surface of CL grows noticeably as the thickness of CL increases. We understood that because the oxygen diffusion was disturbed with water generated by the cell reaction that was accumulated by these cracks, the diffusion overpotential rose, and consequently cell deterioration was caused. Therefore, the dispersion of ionomer should be improved when the CL is applied by IJC.

In the influence of the electrode area on the cell performance, we confirmed that the activation and the resistance overpotentials are hardly influenced by the increase of electrode area. However, because the distance in which the generation steam is multiplied becomes longer with the increase of the electrode area, the diffusion overpotential deteriorates by the plugging phenomena. That is, the condensate is accumulated toward the gas outlet, and causes the plugging by which the condensate blockades the gas passage in the vicinity of the gas outlet. Consequently, the diffusion overpotential of the cell with a large electrode increases.

From these study results, although the effectiveness of IJC was able to be confirmed, improvement of the homogeneous dispersion of ionomer and platinum in catalyst ink and the crack prevention at dryness are necessary.

## Figures and Tables

**Figure 1 polymers-13-00899-f001:**
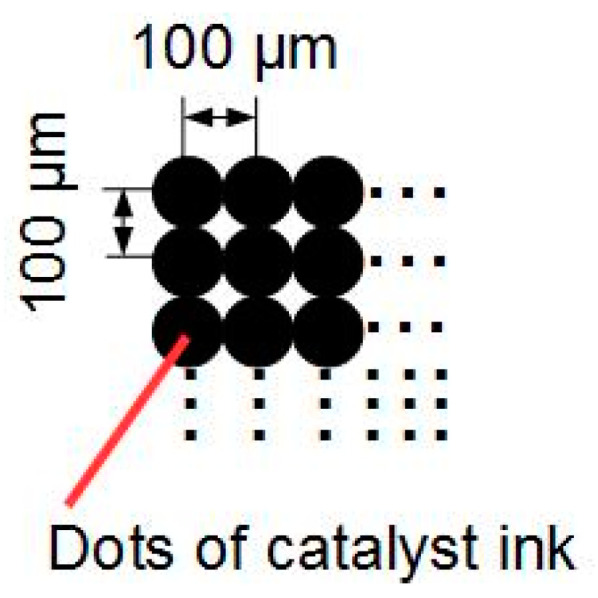
Printing image of catalyst ink on an MPL.

**Figure 2 polymers-13-00899-f002:**
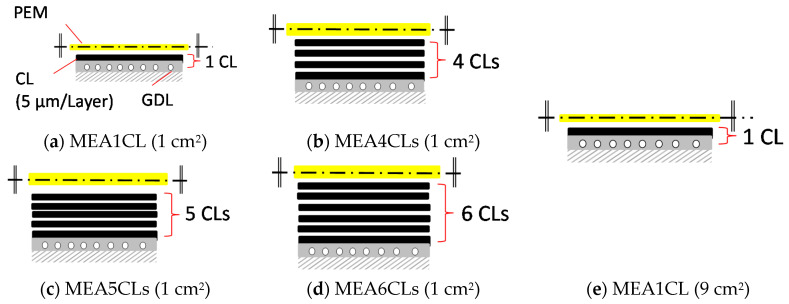
Schematic diagrams of the structure of MEA; (**a**) one CL and 1 cm^2^ of electrode area, (**b**) four CLs and 1 cm^2^ of electrode area, (**c**) five CLs and 1 cm^2^ of electrode area, (**d**) six CLs and 1 cm^2^ of electrode area, (**e**) one CL and 9 cm^2^ of electrode area.

**Figure 3 polymers-13-00899-f003:**
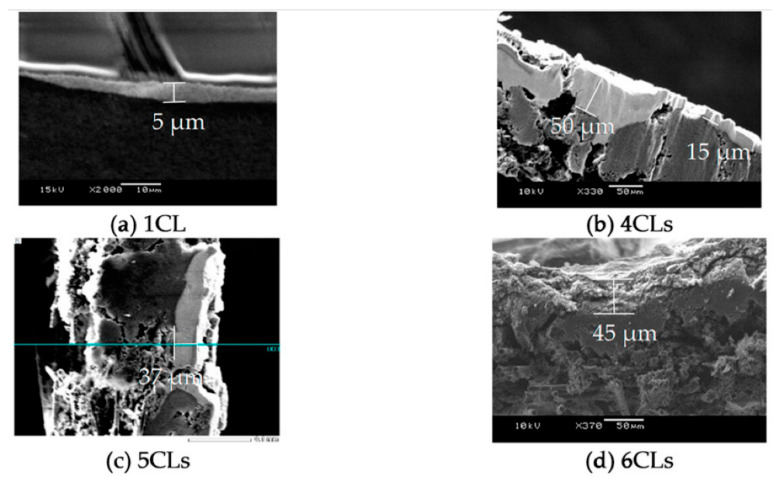
Comparison of SEM image of the cross-sectional area of each CL; (**a**) 1CL, (**b**) 4CLs, (**c**) 5CLs and (**d**) 6CLs.

**Figure 4 polymers-13-00899-f004:**
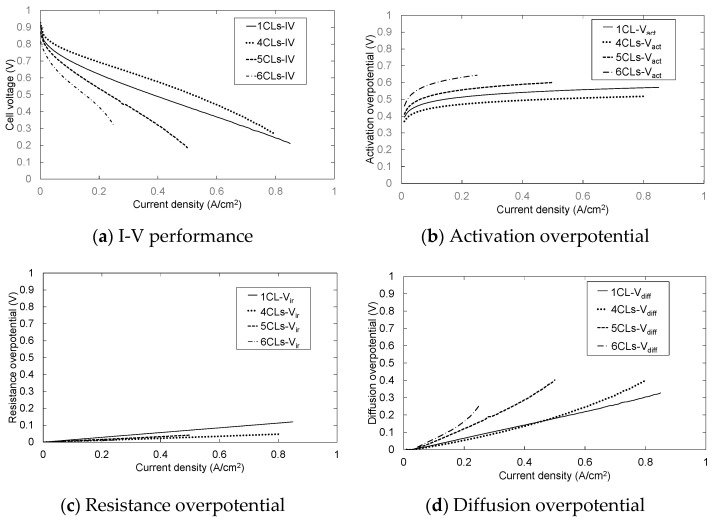
Influence of the number of layers on (**a**) I-V performance, (**b**) activation overpotential, (**c**) resistance overpotential and (**d**) diffusion overpotential.

**Figure 5 polymers-13-00899-f005:**
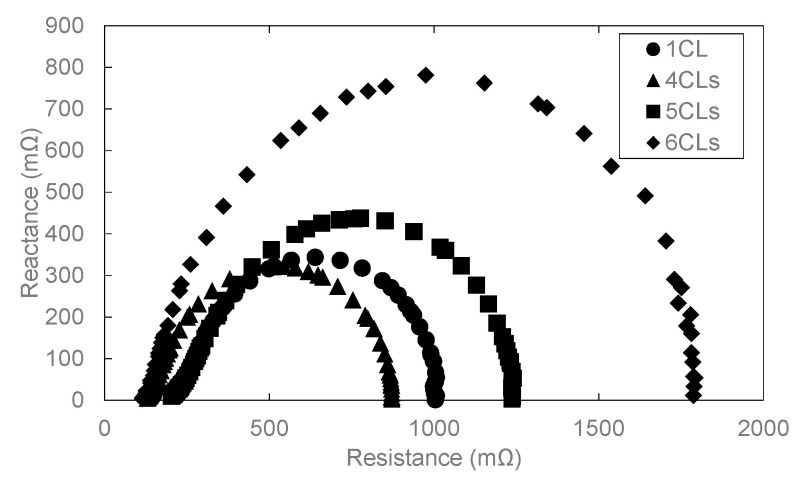
Influence of the number of layers on Cole–Cole plots under 0.1 A/cm^2^ of current density.

**Figure 6 polymers-13-00899-f006:**
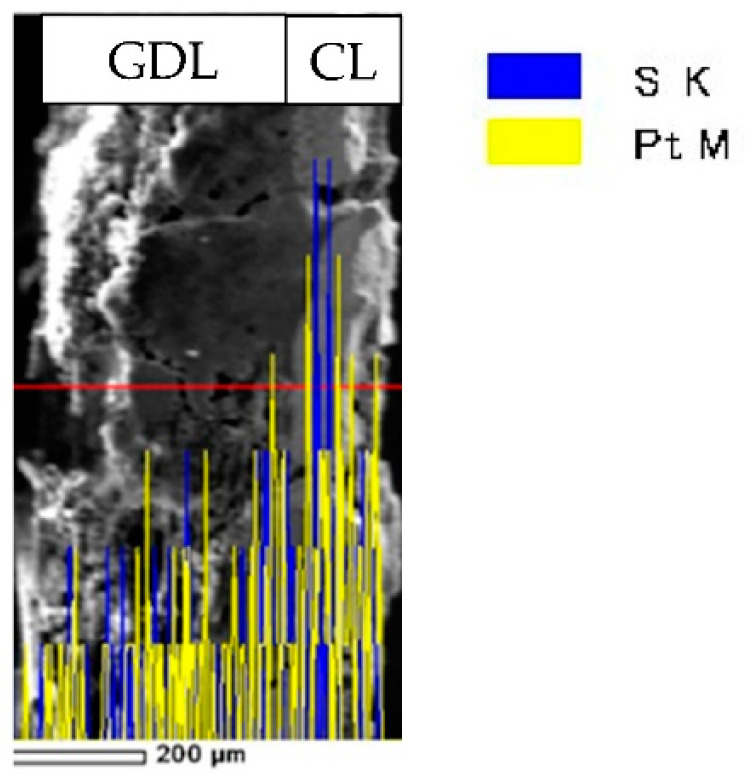
SEM image and EDX analysis on the cross-section of MEA5CLs.

**Figure 7 polymers-13-00899-f007:**
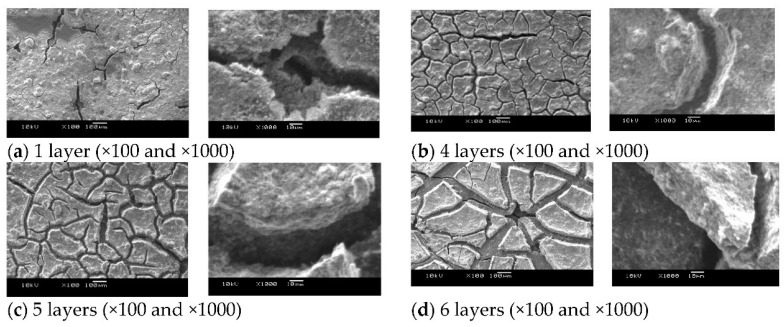
Comparison of SEM image of the surface of each CL.

**Figure 8 polymers-13-00899-f008:**
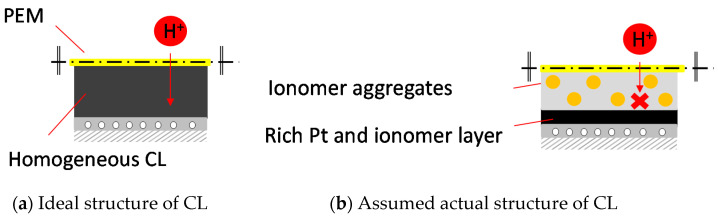
Schematic diagram of influence of number of layers on CL structure.

**Figure 9 polymers-13-00899-f009:**
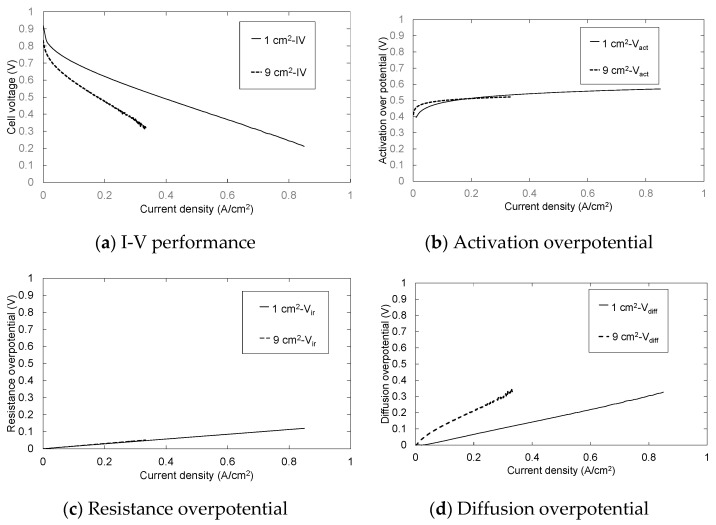
Influence of the electrode area on (**a**) I-V performance, (**b**) activation overpotential, (**c**) resistance overpotential and (**d**) diffusion overpotential.

**Figure 10 polymers-13-00899-f010:**
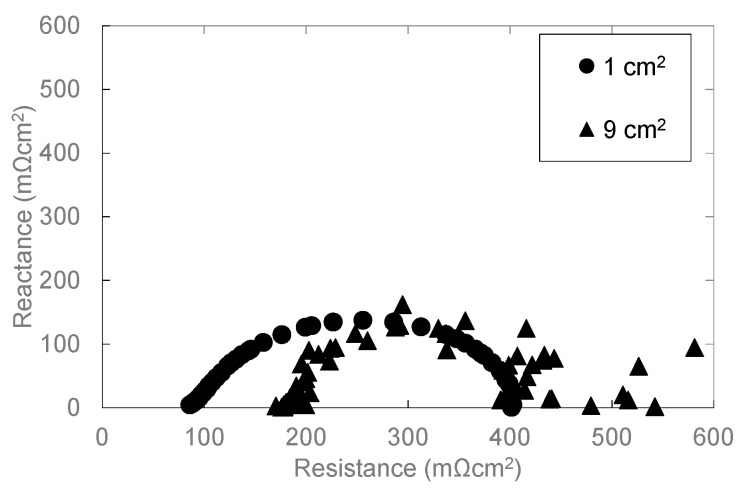
Influence of the electrode area on Cole–Cole plot under 0.1 A/cm^2^ of current density.

**Figure 11 polymers-13-00899-f011:**
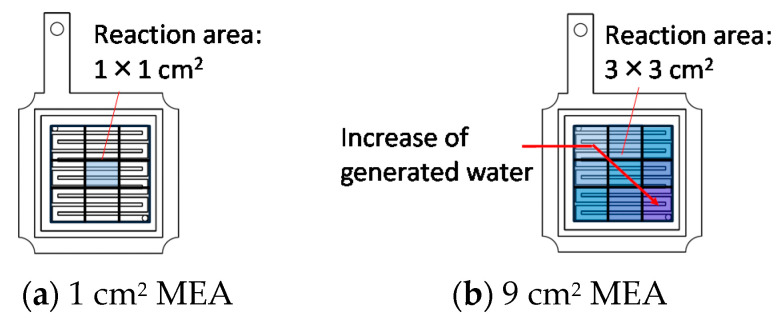
Schematic diagram of influence of the electrode area on drainage. 1 cm^2^ MEA contacts with 0.54 cm^2^ gas channel (**a**), and 9 cm^2^ MEA contacts with 6.12 cm^2^ gas channel (**b**).

**Table 1 polymers-13-00899-t001:** The setting value on the nozzle head driving parameter.

	Setting Values
Applied voltage (V)	30–40
1st pulse width (μs)	100
Pulse interval (μs)	0
2nd pulse width (μs)	0
Frequency (Hz)	100
Droplet speed (m/s)	6.5

**Table 2 polymers-13-00899-t002:** Experimental conditions.

	Setting Values
Standard current density (A/cm^2^)	0.4
Quantity (H_2_/Air) (mL/min)	38.4/160.2
Gas temperature (°C)	80
Gas humidity (%Rh)	100
Cell temperature (°C)	80
Operating pressure of H_2_ gas (hPa)	1500
Operating pressure of Air gas(hPa)	1500
Back pressure (hPa)	1013
Range of impedance frequency (Hz)	0.07–10,000
Superimposed current (mA)	16.5

**Table 3 polymers-13-00899-t003:** Comparison of catalyst ink component.

(wt.%)	Reference [25]	In This Study
Pt/Carbon	6.6	4.2
Nafion	3.4	2.6
Water	60.3	63.8
1-propanol	6.9	5.7
Ethanol	22.8	23.6
Total	100	99.9

## Data Availability

The data are available from the corresponding author on reasonable request.

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
