# Peer review of "Influence of the Catalyst Layer Structure Formed by Inkjet Coating Printer on PEFC Performance"

_polymers, 2021, doi:10.3390/polym13060899_

Round 1
Reviewer 1 Report
Summary:
The manuscript polymers-1143014 titled, “Influence of the Catalyst Layer Structure formed Inkjet Coating Printer on PEFC performance,” report a study of the effect of catalyst layer’s structure toward the performance of a polymer electrolyte fuel cell. Four MEA samples are prepared with 1, 4, 5, and 6 layers of the catalyst. The samples tell that overpotentials of an activation and a diffusion increase with the increase of the thickness of catalyst layer, which also show the single layer MEA with the best performance as compared to other testing samples.
General comment:
In general, this is an interesting work. A minor revision is therefore suggested before the publication.
Comments:
(1) In section 1. Introduction, a combined paragraph is given in the current submission. This makes the motivation and mythology of this work hard to be found. It is suggested to reorganize the introduction section.
Please revise the 1. Introduction by starting “However, the coverage of PEFC…” as the second paragraph in the introduction to tell the motivation and by starting “As one of the methods…” as the third paragraph in the introduction to tell the mythology.
(2) The term “Figure” and “Fig.” are randomly used.
Please revise the manuscript by keeping Figure or Fig. by following the author guideline of Polymers.
(3) From the SEM images, it seems to tell the poor performance of the thicker sample that results from the crack of the CL coating. Once the cracks are solved by additional engineering method, future work might be needed. Please give some comments.
(4) Figure 2 tells the detailed MEA preparation. It would be very nice to see the corresponding cross-sectional SEM image of these four MEA samples since the structure is the main point of this study.
Please provide the cross-sectional SEM images of the four MEA samples.
Author Response
Dear Editor
We have revised our manuscript following your report. Please see the attachment.
Response 1: We have revised the paragraph structure following your comments.
Response 2: We unified the term as “Figure X” in the text and the figure caption.
Response 3: We consider that if the crack is suppressed in laminated CL, the cell performance will improve because it has positive effects on an activation overpotential or diffusion overpotential. However, the issue of ionomer aggregation remains, and we need to future work about it.
Response 4: We have added the cross-sectional SEM images of the four MEA samples on Figure 3 and discussed the thickness at line number 84-90 in our manuscript.
Best regards.

Reviewer 2 Report
The authors reported that the catalyst layer structure's influence formed an inkjet coating printer on PEFC performances. The detailed characterizations studies of (SEM imaging analysis) indicate the nanocomposite material's mechanical property and morphology. The experimental measurements of single-cell performances and impedances of prepared MEA were extensively investigated. As expected, the enhanced, practical and cost-effective MEA have been developed. Overall, this work is interesting, but some minor issues should be addressed before it is published in "Polymers".
This manuscript could be improved by addressing the following issues.
- The author should recheck the typo errors and grammatical errors in the whole manuscript.
- The author can provide some high magnification SEM images for catalysts materials and polymers too.
- The author can justify how the catalyst coating is coated as expected layers of thickness; the author can discuss any catalyst thickness method.
- The introduction part should be informed about recent issues and overcome by the current work in elaborative ways. So, the author should include more points in the introduction part.
- The author could include post experiments analysis after conducting a reliable durability test.
- Several closely related references for this graphene/CNT-based work on the introduction part should be introduced to the broader readers, so it provides more information and enhance the quality and impact of the manuscript., such as 1021/acsami.9b18059; 10.1021/acssuschemeng.9b01757; 10.1016/j.compositesb.2020.107890; 10.3390/polym12091871
Author Response
Dear Reviewer
We have revised our manuscript following your report. Please see the attachment.
Response 1: We have rechecked the typo errors and grammatical errors. However, we ignored words that Word program misidentified as a misspelling such as product name, Person’s name and other proprietary nouns.
Response 2: We added high magnification SEM images on Figure 7.
Response 3: We have added the cross-sectional SEM images of the four MEA samples on Figure 3 and discussed the thickness at line number 84-90 in our manuscript.
Response 4: We informed about recent issues of PEFC and IJP method at the second and third paragraph respectively. We added and informed the necessity of this research at line number 63,64.
Response 5: We are going to conduct durability tests based on Japan Automobile Research Institute regulations. However, it is difficult to include the reliable results in this paper until deadline because the examination takes a long time, in addition, it is necessary to confirm the reproducibility.
Response 6: We added two references you provided (10.1016/j.compositesb.2020.107890; 10.3390/polym12091871) in the introduction. Along with this revise, the text at line number 35-43 was changed.
Best regards.
